# Combining Transurethral Resection of Fibrous Tissue and Temporary Urethral Stent Insertion Is an Optimal Strategy for Minimally Invasive Treatment of Recurrent and Long Urethral Strictures

**DOI:** 10.3390/jcm12051741

**Published:** 2023-02-22

**Authors:** Sun-Tae Ahn, Seon-Beom Jo, Hyun-Soo Lee, Du-Geon Moon

**Affiliations:** Department of Urology, Korea University Guro Hospital, No. 148, Gurodong-ro, Guro-gu, Seoul 08308, Republic of Korea

**Keywords:** urethra, chronic, stricture, fibrotic tissue, stenting, transurethral remove

## Abstract

This study investigated the optimal strategy for the treatment of chronic recurrent urethral strictures longer than 3 cm, using a temporary urethral stent. Between September 2011 and June 2021, 36 patients with chronic bulbomembranous urethral strictures underwent temporary urethral stent placement. Retrievable self-expandable polymer-coated bulbar urethral stents (BUSs) were placed in 21 patients (group A), and thermo-expandable nickel-titanium alloy urethral stents were placed in 15 patients (group M). Each group was subdivided into those with and without transurethral resection (TUR) of fibrotic scar tissue. The urethral patency rates at 1 year after stent removal were compared between the groups. The patients in group A showed a higher urethral patency maintenance rate at 1 year after stent removal than those in group M (81.0% vs. 40.0%, log rank test *p* = 0.012). Analysis of subgroups in which TUR was performed due to severe fibrotic scar, showed that the patients in group A showed a significantly higher patency rate than patients in group M (90.9% vs. 44.4%, log rank test *p* = 0.028). In the treatment of chronic urethral strictures with a long fibrotic scar, temporary BUS combined with TUR of fibrotic tissue seems to be the optimal minimally invasive treatment strategy.

## 1. Introduction

Urethral stricture, especially due to recurrent long fibrotic scar tissue, is one of the most intractable urological diseases. Although urethroplasty is the gold standard for the treatment of focal bulbar strictures (<1 cm) with a high success rate [1], variable success rates have been reported for this procedure in redo-urethroplasty, posterior strictures, and longer strictures [2,3]. For the development of a minimally invasive treatment for urethral stricture, direct visual internal urethrotomy (DVIU) and urethral dilatation are applied as first-line treatment; however, poor results have been reported in cases where the stricture is longer than 1 cm [4], with repeated procedures and more complications where the stricture is longer than 2 cm [4,5].

For the last 30 years, urethral stents have been developed to increase the efficacy of minimally invasive treatment for urethral strictures, but permanent stents have been withdrawn from the market [6]. Permanent use and the absence of a coating system for the metallic coil are responsible for encrustations, stone formation, tissue ingrowth through the stent leading to obstruction, and the need for surgical removal of the stent [7,8]. Recently, favorable results of temporary urethral stent insertion for the management of urethral strictures have been reported [9,10]. However, most of the studies placed the stent after DVIU/dilation and maintained 3 to 12 months of stent time in cases where the stricture length was less than 2.5 cm [10,11,12,13].

Fibrosis is the most important underlying process for the development and recurrence of urethral strictures following urothelial injury [2]. The authors have already suggested the necessity and feasibility of endoscopic removal of fibrotic scar tissue and demonstrated the efficacy of a temporary urethral stent in traumatic urethral rupture [14,15,16]. Besides various factors such as the length of urethral stricture, type of stent, and stent indwelling period, complete removal of fibrotic scar tissue is an important factor for stable placement of the stent and regeneration of normal urethra to prevent the recurrence of stricture. However, previous reports did not apply these factors to treatment due to concerns about additional urethral damage. In this study, we assessed the long-term outcome of a temporary urethral stent after DVIU with additional transurethral resection (TUR) of fibrotic scar tissue in a chronic recurrent urethral stricture longer than 3 cm.

## 2. Materials and Methods

### 2.1. Patients and Study Design

This retrospective cohort study included patients who underwent DVIU with or without TUR of fibrotic urethral tissue and temporary urethral stent insertion (self-expandable stent or thermo-expandable stent) for recurrent long urethral strictures. The data of the self-expandable stent were compared with the retrospective data of the thermo-expandable stent. Institutional review board (IRB) approval was obtained for this study (IRB No. 2020GR0006).

We analyzed patients who had recurrent bulbar strictures between September 2011 and June 2021. All patients were diagnosed with urethral stricture based on their medical history, maximal uroflow on uroflowmetry, urethroscopy, and retrograde urethrography. Recurrent long urethral stricture was defined as a narrow stricture segment of >3 cm on urethrography and/or urethroscopy and weak uroflow, a Qmax of <5 mL/s, and a history of failed DVIU more than two times.

Patients who underwent attempts at catheterization before retrograde urethrography were also excluded, as were patients with a follow-up duration of less than 1 year after stent removal. Ultimately, 36 patients were included in this study.

The patients were divided into two groups according to the type of temporary stent used. The patients who received thermo-expandable stents were defined as group M, and the patients who had received a self-expandable stent were defined as group A. Each group was subdivided into two groups: patients who had received DVIU and dilatation before stenting were defined as groups AD or MD, and the patients who had received additional endoscopic removal of fibrous tissues were defined as groups AT or MT. The patients’ flow-chart is summarized in Figure 1. The baseline characteristics, including age, urethral stricture length, and stent indwelling duration, were compared between the two groups.

### 2.2. Stents

Two types of stents were used in this study. Thermo-expandable urethral stents (Memokath 044TW, Pnn Medical, Kvistgaard, Denmark) were used in group M, and self-expandable stents (Allium Bulbar Urethral Stent, Allium LTD, Caesarea, Israel) were used in group A. The Memokath stent is made of a tightly coiled nickel–titanium alloy that is designed to prevent urothelial ingrowth. It has a thermosensitive shape memory that expands either at the proximal segment or at both ends of the stent from 24 to 42 Fr using warm water (55 °C) instillation to facilitate the anchoring of the stent in the correct position. When a cooled irrigant is used (approximately 5–10 °C), the expanding ends become soft and pliable, allowing for easy removal. The stents ranged in length from 3 to 8 cm in 1 cm increments, which allowed selection of the appropriate stent length for individual patients.

The Allium bulbar urethral stent (BUS) is a fully covered, large-caliber metal stent, specially designed for the treatment of bulbar urethral strictures (Figure 2). The stent is built of a coiled, superelastic metal alloy (nitinol) covered with a polymeric coating designed to prevent mucosal hyperplasia and encrustation. The main body (Figure 2B) acts as a mold to allow the formation of a large urethral lumen expanding to a 45 Fr caliber. The dynamic sphincteric segment (Figure 2A) prevents sphincteric dysfunction that may lead to incontinence. The last portion of the stent is the soft distal segment (Figure 2C). A special feature allows stent retrieval by unraveling it into a thread-like strip, which enables a nontraumatic removal. We used three different lengths: 60, 80, and R 80 mm.

### 2.3. Management Strategy and Assessments

Under general anesthesia, the patients were placed in the lithotomy position. Under endoscopic guidance, a 20 Fr urethrotome was introduced into the urethra. Multiple incisions at different positions were made using a cold knife to achieve a minimum inner diameter of 26 Fr for advancement of the 22 Fr delivery system and minimum expansion of the 26 Fr expandable stent. In patients with severe fibrotic stricture or a constriction ring from traumatic contracture, fibrotic rings and fibrotic scars were completely removed by cutting with a 24 Fr TUR loop and achieving enough diameter for full expansion of the urethral stent. After complete removal of fibrotic tissues and achievement of proper caliber of the urethra, the stricture length was measured. Subsequently, either a thermo-expandable nickel–titanium alloy urethral stent or a self-expandable polymer-coated BUS stent was deployed with a specially designed delivery system to cover the stricture site along with an extra 1 cm at both ends. A Foley’s catheter was not inserted.

Patients were advised to visit the outpatient clinic at 1, 3, 6, 9, and 12 months postoperatively. During follow-up, uroflowmetry, pain or dysuria, ejaculatory disturbance, and urethroscopy were assessed in patients with Qmax < 10 mL/s to find any granulation tissues or migratory obstruction. The stent was removed 3–12 months after the operation depending on the presence of granulation tissue at the stent edge and patient compliance. After stent removal, we explored the previous stricture site using a ureteroscope.

All patients were instructed to return for follow-up at 1, 3, 6, and 12 months after stent removal. All the patients were assessed using uroflowmetry and urethroscopy at each visit. If a weak stream developed immediately after stent removal, periodic dilation was applied until tolerable uroflow was achieved with Qmax > 10 mL/s. During the initial follow-up after stent removal, all patients who showed urethral overgrowth at the edge of the stent insertion site underwent urethral dilatation.

The clinical success rate and complications were compared between groups M and A during both the stent indwelling period and after stent removal. Clinical success was defined as the maintenance of urethral patency in urethroscopy, passing of a 17 Fr cystoscope without resistance, and a maximum urinary flow rate of >10 mL/s without requiring additional surgery, such as visual internal urethrotomy or urethroplasty. In both groups, a sub-analysis was performed using DVIU and DVIU +TUR. Stent-related complications, including migration, discomfort, tissue ingrowth, granulation at the stent edge, stone formation, and post-void dribbling, were recorded and compared between the groups.

The Mann–Whitney U test was used to compare groups for continuous variables, and Fisher’s exact test was used to compare categorical variables. A two-sided *p*-value < 0.05 was considered statistically significant for all tests. Kaplan–Meier curves were used to determine the success rate over 12 months. Differences in the success rate according to the stent type and whether TUR of the urethral fibrous scar tissue was performed, were assessed using a log-rank test. All analyses were performed using IBM SPSS Statistics version 22.0 (IBM Co., Armonk, NY, USA).

## 3. Results

In this study, 21 patients were treated with self-expandable stents and were classified as group A. For the control group, the medical records of 15 patients treated with thermo5-expandable stents were classified as group M. The demographic and clinical characteristics of the two groups are presented in Table 1. There were no significant differences between groups M and A in terms of the mean patient age and urethral defect length. Due to the differences in stent characteristics and separate retrospective data, the mean duration of stent placement was significantly longer in group M (8.5 months) than in group A (4.8 months) (*p* = 0.008).

In both groups, granulation tissue at both edges of the stent and stone formation were seen as foreign body reactions. The granulation tissue formation rate at the stent edge differed significantly (*p* = 0.011) between the groups, with 73.3% patients in group M and 38.1% patients in group A. Tissue granulation at the stent edge was managed by repeat urethral dilatation during the first 3 months after stent removal. Only one patient in group M required transurethral resection of granulation tissue due to sustained obstructive granulation tissue, even after repeated urethral dilatations. Post-void dribbling was the most frequent complication during stent indwelling in group M (12/15, 80.0%), whereas only four patients (4/21, 19.1%) in group A demonstrated symptoms (*p* = 0.001); post-void dribbling was controlled by Kegel exercises with or without an anticholinergic drug. In group M, stent-related pain or discomfort was high but was controlled with analgesics until the stent was removed. Other stent-related complications, including stent migration and encrustation, did not differ significantly between the groups.

The overall incidence of urethral patency at 1 year after stent removal was 81% in group A, which was significantly higher than 40% in group M (Figure 3). In the subgroup analysis, there was no significant difference according to the stent type in the MD and AD groups, in which only DVIU was performed without TUR (Figure 4). However, in the comparative analysis of groups MT and AT, in which TUR was performed due to a severe fibrotic scar or ring, there was a significant difference in the success rate 1 year after stent removal depending on the stent type (group MT, 44.4% vs. group AT, 90.9%; log-rank test *p* = 0.028) (Figure 3). Three patients who failed to maintain urethral patency after stent removal in group M were switched to Allium BUS stents. Two patients (9.5%) in group A were switched to CIC and suprapubic cystostomy for undiagnosed detrusor failure. None of the patients required a repeat DVIU or open surgical urethroplasty.

## 4. Discussion

The main treatments for recurrent urethral strictures are open urethroplasty, DIVU, urethral dilation, and urethral stenting. The treatment of urethral strictures depends on the length, location, and type of stricture [2]. The treatment principles of open urethroplasty, excision, and end-to-end anastomosis are complete removal of fibrotic tissue, including a fibrotic restrictive ring, which prevents recurrent fibrosis and allows tension-free anastomosis. In both open urethroplasty and minimally invasive temporary stenting, the length of the urethral stricture is one of the most important factors for successful treatment. Instead of excision, which cannot be applied in urethral strictures longer than 3 cm, buccal mucosal onlay grafts can be applied with a success rate of 86% even in long strictures (mean 4.6 cm) [17]. However, a longer urethral stricture is a real challenge for minimally invasive treatment using a temporary urethral stent. This study was performed to assess the efficacy of temporary urethral stenting in chronic recurrent urethral strictures >3 cm (mean, 4.8 cm). According to the main principle of open urethroplasty, complete removal of fibrotic tissue, including the fibrotic restrictive ring, prevents recurrent fibrosis.

In this context, the current study aimed to assess the efficacy of TUR in removing fibrotic tissue to achieve a better environment for regeneration of new urethral tissue. This would enable the stable placement of temporary urethral stents with good patency until the urethral mucosa is stabilized. The causes of the lower success rates in treating long urethral strictures are: difficulties in complete removal of long fibrotic segments and achievement of a fibrosis-free, large-diameter urethra capable of accepting fully-expanded stents [18]. Temporary stents after DVIU showed lower success rates for strictures longer than 1 cm [10], which might be the result of incomplete expansion of the expandable stent or the restrictive force of fibrotic tissue against the radial force of the stent. DVIU and urethral dilation within the fibrotic scar cause temporary widening and lead to eventual recurrence with repeated procedures. In patients with long recurrent strictures, DVIU alone is not sufficient to achieve proper caliber of the urethra for full expansion of expandable stents, especially for long narrow fibrotic scars or thick fibrotic restrictive rings in the urethra. The main drawbacks of DVIU or urethral dilation are the persistence of fibrotic tissue and immediate recurrence of the block after stent removal, even with a long stent time.

The authors assumed that removal of the fibrotic ring or segment could offer a suffi-ciently expandable caliber of the urethra. Endoscopic removal of fibrotic scar tissue decreases the recurrence of the growth of the remaining scar tissue and enhances regeneration of normal urethral tissue. First, transurethral endoscopic removal of the fibrospongiosis tissue effectively nullifies any source of recurrence. Second, transurethral endoscopic removal of fibrospongiosis exposes the underlying normal tissue layer as a cell source necessary for the regeneration of normal urethral mucosa. Third, transurethral endoscopic removal of fibrotic contracture or deep fibrotic scar tissue can achieve a sufficiently widened urethra compatible with a fully expanded stent.

For the reconstruction or regeneration of tubular structures, such as vessels, ureters, or urethrae, patent scaffolds and regenerating endothelial or urothelial cells are essential. For the regeneration of normal urethra from urethral strictures, scar-free widened urethra can be supported by various urethral stents as temporary mechanical supports or tubular splints, not as biodegradable scaffolds. Wong et al. [19] hypothesized that temporary urethral stents might aid in the management of recurrent urethral strictures. The temporary stent could act as a scaffold to splint against the mechanical forces of scar tissue contraction during the healing period, and this action may ultimately stabilize the stricture site during epithelization [19]. Therefore, the need for further endoscopic or urethroplasty procedures has reduced.

In the current study, the Allium BUS stent group was compared with the Memokath stent group using retrospective medical records. Despite heterogeneous data, the same protocol was followed in both groups; however, the success rate was significantly higher in the Allium BUS group than in the Memokath group. In a total analysis of our study with previous study data, the Memokath stent showed a success rate of 40.0% (6 of 15) for recurrent long strictures. The Allium BUS stent showed a higher success rate of 81.0% (17 of 21) for recurrent strictures. In patients with TUR, the failure rate was higher for Memokath than for Allium BUS. Group A showed a significantly higher Qmax until 1 year after stent removal. The Allium BUS stent was larger than the Memokath stent (15 mm vs. 8 mm). Additionally, the Allium BUS stent had the same diameter of 45 Fr (15 mm) from the body to one end, and the other end was tapered as a soft conical structure to fit the external sphincter. Memokath stents had three different diameters: 8 mm (24CH) for the stent body, 10 mm (33CH) for the small anchor, and 12 mm (42CH) for the large anchor.

For full expansion of the Allium BUS stent, the whole stricture length should be widened to an inner diameter of 15 mm with DVIU, TUR, and dilation with metal sound. After complete removal of the fibrotic tissue and contracture ring, the Allium BUS stent provided sufficient radial force as resistance to prevent the collapse of the lumen with a large diameter (1.2 cm) until stabilization of the urethra. After removal of the Allium BUS stent, total collapse will not usually occur, and periodic dilation can minimize the recurrence in cases of gradual narrowing that occurs during follow-up. To regenerate the normal stable large-lumen urethra from a recurrent fibrotic long stricture, the initial large lumen of the Allium BUS stent would be expected to yield better results than the small lumen of the Memokath stent after removal.

The Memokath stent luminal dilation up to 12 mm is sufficient for stent placement, but it is unnecessarily large for the stent body. As regards to the diameter of the main stent body, the Memokath stent (8 mm) is smaller than the Allium BUS stent (15 mm). Despite good patency with a proper lumen with Memokath in situ, luminal collapse and resultant recurrence can easily occur after stent removal for small diameters. Compared with the small diameter of the main body, the large diameter of the anchor may induce the growth of granulation tissue at both ends. Moreover, fibrotic tissue grows into the lumen via the coil and luminal obstruction can occur sometimes. For the Allium BUS stent, the outer layer is covered; therefore, tissue ingrowth is prevented. Regardless of the TUR, the success rate was low for MemoKath stents. The Memokath group showed good uroflow with the stent in situ; however, the follow-up result was poor after stent removal. This means that the difference in stent diameter is the main cause of the differences in success rates. In permanent stents, a stent diameter of 8 mm is not small; however, a large caliber is better for temporary stents, especially for long fibrotic strictures. After removal of the stent, the neourethra may collapse due to fibrotic growth or outer constriction. Recurrent strictures will be more prevalent in urethrae with small lumina than in those with large ones. To avoid recurrent stricture after stent removal, the neourethra should be wide, stable, healthy, and free from fibrous scars.

For successful treatment by temporary stents, the stent should remain in the stricture site with full expansion until complete regeneration of the urethra occurs. A large-caliber urethra devoid of fibrotic tissue is essential for the stable placement of Allium BUS stents and resultant successful treatment. A narrow fibrotic urethra induces problems in both types of stents. In the narrow fibrotic urethra, Allium BUS stents do not expand completely, are obstructed by collapse, and are sometimes immediately displaced during urination.

With respect to stent times for various temporary stents in recurrent strictures, a minimum indwelling period of 4 months resulted in less stricture recurrence with a 55% success rate [20], and the UroCoil™ temporary stent for 12 months resulted in an 83% success rate at a 24-month follow-up after removal of the stent [21]. However, it is still not clear whether a shorter or longer stent time achieves better results because it depends on the heterogeneity of patient profiles, different features of the stents, and the experience and expertise of the operator. Short-term stent placement also has the advantage of fewer complications such as migration, discomfort, incontinence, infection, and encrustation. Premature stent removal before complete recovery has been associated with higher rates of recurrence [20]. Üstüner et al. recently reported long-term follow-up results after BUS removal for bulbar urethral strictures [13]. Taken together, a longer indwelling time of between 8 and 18 months (mean, 10.9 months) was significantly (*p* = 0.009) related to a higher clinical success rate than a shorter period of between 3 and 7 months (mean 4.6 months).

In this study, stents were removed at 4 months in group A and at 8 months in group M. Despite the shorter stent indwelling period, group A showed a higher success rate and a significantly higher Qmax at 1 year after removal of stent. In contrast to the Turkish study, we removed fibrotic scars using TUR. Unlike other studies, it was assumed that the TUR group requires a shorter stent period until regeneration of the new urethra. In other studies, the temporary stent required more time to avoid the recurrence of fibrotic strictures and to stabilize the new urethra [19]. In patients with recurrence, a longer stent time could be attempted for better results.

Compared with the recent report on the Allium BUS stent, our study has several limitations. First, we could only analyze data from a small number of patients with a 1-year follow-up. Second, data from the Memokath group were obtained from retrospective medical record reviews and available telephone interviews. Randomization could not be achieved, and the study results may have been affected by selection bias. Third, we did not compare our treatment strategy with that of buccal mucosal urethroplasty. However, our study was primarily focused on the endoscopic removal of scars to ensure meaningful reduction of the recurrence of strictures. Fourth, sexual dysfunction was not evaluated in this study. Due to its retrospective design, baseline sexual function was unknown in the study cohort of this study and could not be evaluated appropriately. Despite these limitations, we believe that this study prompts further investigation into the improvement of minimally invasive treatments for chronic recurrent long-segment urethral strictures.

## 5. Conclusions

Previous treatment modalities for the management of chronic recurrent urethral strictures with a long fibrotic segment were unable to reduce the recurrence of fibrosis. Urethral stents have long been used to preserve luminal patency, and with the development of endoscopic procedures, these stents have been further developed for various purposes. The majority of previous studies have focused on revolutionary urethral stents and ideal stent periods after DVIU for strictures less than 2.5 cm. In this study, we investigated temporary urethral stents with or without complete removal of fibrosis using TUR in patients with long stricture lengths and chronic recurrent strictures. Despite the heterogeneity of both groups, the Allium BUS stent showed better efficacy and lower complication rates than the MemoKath stent in the management of chronic recurrent urethral strictures. Where thick fibrous tissue or constriction rings are present, complete removal of the fibrotic tissue by additional TUR rather than by DVIU is a suitable option for successful regeneration of the urethra.

## Figures and Tables

**Figure 1 jcm-12-01741-f001:**
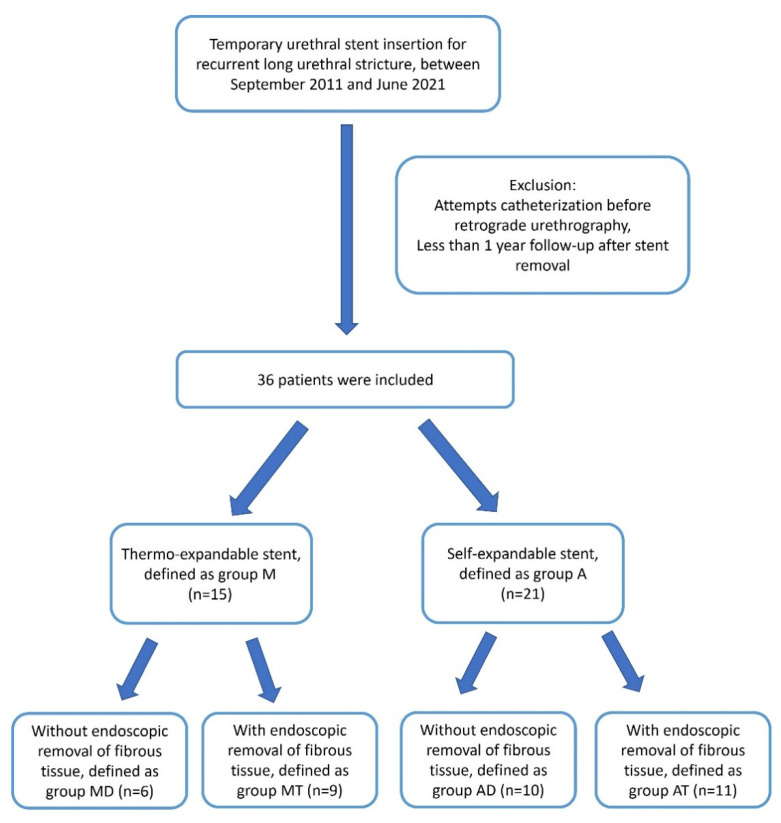
Flow-chart for patient selection and their division into groups.

**Figure 2 jcm-12-01741-f002:**
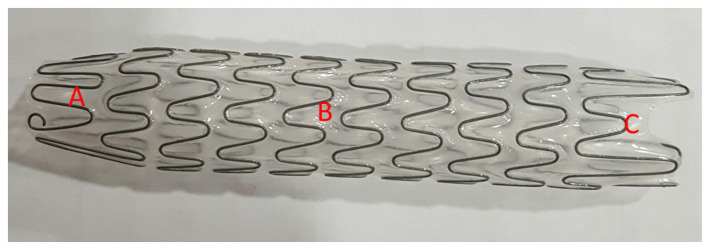
An Allium bulbar urethral stent (BUS). (**A**) Soft sphincteric segment; (**B**) high radial force body; (**C**) soft distal segment.

**Figure 3 jcm-12-01741-f003:**
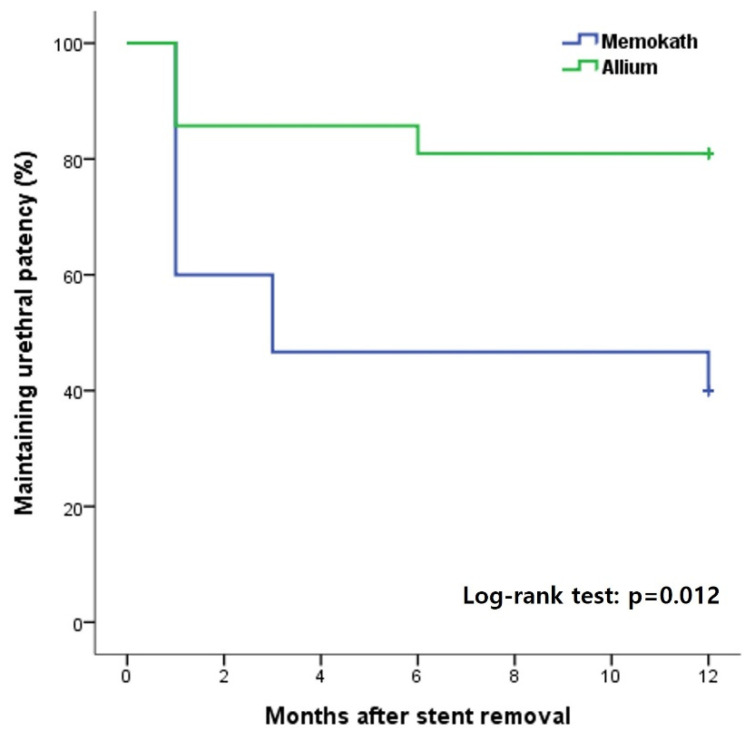
Kaplan–Meier curves for overall success rate by stent type over 12 months after stent removal.

**Figure 4 jcm-12-01741-f004:**
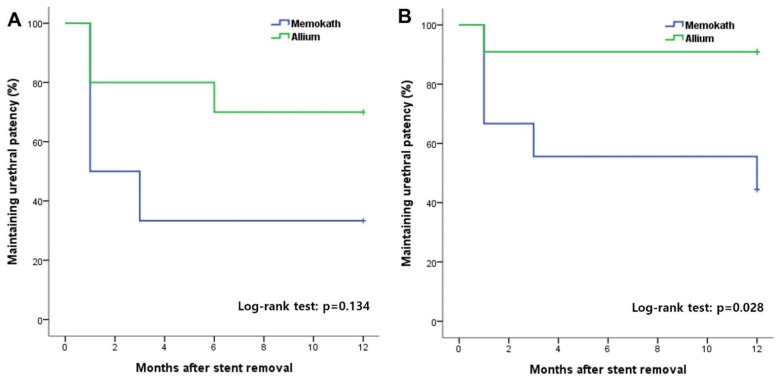
Kaplan–Meier curves showing the maintaining urethral patency rate with Memokath and Allium stents in groups subdivided into patients with direct visual internal urethrotomy with (**B**) or without (**A**) transurethral endoscopic removal of urethral scar tissue.

**Table 1 jcm-12-01741-t001:** Baseline characteristics and outcomes of stent indwelling in both groups.

	Group M (MemoKath)	Group A (Allium BUS)	*p* Value
No Pts	15	21	
Trauma or VIU Hx. (+)	3	6	
Stricture length (cm)	4.5 ± 1.72	5.0 ± 3.71	0.293
Stent period (month)	8.5 ± 4.49	4.8 ± 3.45	0.008
Qmax, stent in situ	14.7 ± 4.93	20.9 ± 6.22	0.008
6M after stent removal	14.0 ± 6.94	24.3 ± 6.22	0.020
Pain or discomfort (%)	10 (66.7%)	2 (9.5%)	0.001
Granulation, stent edge (%)	11 (73.3%)	8 (38.1%)	0.011
Stone formation	3 (20.0%)	6 (28.6%)	0.705
Luminal ingrowth	3 (20.0%)	0 (0)	0.064
Post-void dribbling	12 (80.0%)	4 (19.10%)	0.001
Stent migration	2 (13.3%)	3 (14.3%)	0.663

VIU; visual internal urethrotomy.

## Data Availability

The data are not publicly available due to privacy policy of our institute IRB.

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
