# Peer review of "Combining Transurethral Resection of Fibrous Tissue and Temporary Urethral Stent Insertion Is an Optimal Strategy for Minimally Invasive Treatment of Recurrent and Long Urethral Strictures"

_jcm, 2023, doi:10.3390/jcm12051741_

Round 1

Reviewer 1 Report

This is an interesting trial to encounter the problem of urethral strictures. Although the suggested techniques are widely approved, the combined methods are quite challenging and with lack of published studies. 

However, there are several methodological questions:

- please verify the exact time interval of recruitment (2011 or 2016 till 2021)

- is that a retrospective or a prospective study? Any kind of mixing retro with prospect cases cannot be statistically approved, so you have to define it at the methods section and then analyze the respective sample

- conclusions has to be re-evaluated after the above definition

Author Response

Point 1: please verify the exact time interval of recruitment (2011 or 2016 till 2021)

Response 1: Thank you for pointing out. The patients analyzed in this study were from 2011 to 2021, so the’ Material & methods’ in the manuscript were modified accordingly.

Point 2: is that a retrospective or a prospective study? Any kind of mixing retro with prospect cases cannot be statistically approved, so you have to define it at the methods section and then analyze the respective sample

Response 2: I agree with your logical point. Although the patients who underwent self-expandable stent in this study were recruited prospectively, the control group for comparison was patients who had previously undergone thermos-expandable stent, so as you point out, this study was applicated as a retrospective study. Therefore, we clarified in material methods that this study was a retrospective study.

Point 3: conclusions has to be re-evaluated after the above definition

Response 3: I agree with your point. In conclusion, the sentence, “Both temporary stents were effective in preserving urethral patency during the long-term follow-up.”, has been deleted because it was contradictory to the results.

Reviewer 2 Report

The authors have performed a retrospective study comparing Memokath vs Allium. The conclusions of this study have to be mitigated. The authors demonstrate that both stents are inadequate for managing urethral strictures and require removal after about 6 months. Urethral stents are not considered a recommended treatment modality due to their high rate of complications and their short time to explantation. Comparing two non-recommended stents with each other does not aid clinical practice.

Author Response

Thank you for your review comments. In principle, both stents are temporarily maintained and removed, not permanently. The stent is maintained until the complete regeneration of the urethra is completed, and the removal was the presence of granulation tissue at the stent edge and patient compliance, as stated in the text.  As described in the revised conclusion, the self-expandable stent was more effective than the thermo-expandable stent among the two stents and was maintained significantly that performed fibrous tissue removal in the sub-analysis.

Reviewer 3 Report

1) In Abstract you reported that you enrolled the patients between September 2011 and June 2021, but in Materials and Methods you reported "between Nov 2016 and June 2021".

2) I think that a flow-chart for the patient selection and their division into groups

3) What other stent-related complications did you evaluated? Why did you not included these complications in Table One?

4) Can you explain the abbreviation in Table One?

5) Do you have some references for lines 213 - 236?

Author Response

Point 1: In Abstract you reported that you enrolled the patients between September 2011 and June 2021, but in Materials and Methods you reported "between Nov 2016 and June 2021". 

Response 1: Thank you for pointing out. The patients analysed in this study were from 2011 to 2021, so the’ Material & Methods’ in the manuscript were modified accordingly. 

Point 2: I think that a flow-chart for the patient selection and their division into groups

Response 2: Thank you for comments. We added patients flow-chart.

Point 3: What other stent-related complications did you evaluated? Why did you not included these complications in Table One?

Response 3: Thank you for comments. We added data for stent migration to table 1.

Point 4: Can you explain the abbreviation in Table One?

Response 4: Thank you for comments. We added an explanation of abbreviation below the table.

Point 5: Do you have some references for lines 213 - 236?

Response 5: Thank you for comments. We added reference for lines 213-216.

Round 2

Reviewer 1 Report

No more comments. Thank you for the definitions.

Author Response

Thank you for completing revision. 

Reviewer 2 Report

I summarized my comments in the previous review

Author Response

Thank you for reviewing our paper and your valuable comments.

Reviewer 3 Report

The Authors followed Reviewers' suggestions and the manuscript is now more complete, but I think that you the number of patients for every group and subgroup should be added in Figure 1

Author Response

Point 1: The Authors followed Reviewers' suggestions and the manuscript is now more complete, but I think that you the number of patients for every group and subgroup should be added in Figure 1

Response 1: Thank you for comments. We added the number of patients for every group and subgroup om Figure 1.
